# A Thematic Synthesis Considering the Factors which Influence Multiple Sclerosis Related Fatigue during Physical Activity

**DOI:** 10.3390/bs9070070

**Published:** 2019-07-01

**Authors:** Sofia Mezini, Andrew Soundy

**Affiliations:** 1School of Sport, Exercise & Rehabilitation Sciences, University of Birmingham, Edgbaston, Birmingham B15 2TT, UK; 2Stratford Hospital, Arden Street, Stratford, Warwickshire CV37 6NX, UK

**Keywords:** multiple sclerosis, fatigue, exercise, physical activity, experience, qualitative, meta-synthesis

## Abstract

The purpose of this study is to consider the factors that influence fatigue related to physical activity in patients with multiple sclerosis (PwMS) and to identify the necessary adaptations undertaken by patients to remain active. A review using a thematic synthesis methodology situated within a subtle realist paradigm was undertaken. The review was completed in three stages: 1) search of relevant studies; 2) critical appraisal of literature; and 3) thematic synthesis. Nineteen studies met the inclusion criteria. This included a total of 263 participants of whom 243 were PwMS (159 females, 70 males and 14 unknown). The aggregated mean age was 53.3 years and aggregated mean time living with MS post diagnosis 11.3 years. Following critical appraisal, no articles were excluded. Three major themes were identified: (1) fatigue-related consequences, (2) exercise related barriers affecting fatigue, and (3) factors that make fatigue bearable for MS individuals. The thematic synthesis identified the cycle of activity and inactivity as a result of fatigue perception. Exercise experience, professional and social support, as well as the necessary adaptation of a training programme empower PwMS to adopt a more active coping strategy and enjoy the benefits of exercise. Clinicians could consider the implementation of a suitable, individualised exercise programme to reduce PwMS’s stress during physical activities.

## 1. Introduction

Multiple sclerosis (MS) is a chronic progressive demyelinating disease of the central nervous system (CNS) [1,2]. It causes neurological disability in young adults with the peak onset age between 40–50 years [3]. MS affects approximately 2.5 million patients worldwide [1]. This represents a prevalence of around 0.02%–0.2% in Europe [2,4]. This includes a higher prevalence in Scotland and Northern Ireland [2,4] as well as some populations in Scandinavia [2]. 

Approximately 80% of patients with multiple sclerosis (PwMS) have a relapsing-remitting (RR) type, which is characterised by acute episodes of neurologic dysfunction (relapses), followed by partial or full recovery. MS is more prevalent in females than males by a ratio of 2:1 [5,6]. The remaining 10%–20% of PwMS have a primary or secondary progressive type, which affects both genders equally and is characterised by progressive neurological deterioration and increasing neurological disability [5,6].

Fatigue is the most commonly reported symptom among PwMS and more frequent in progressive than in RR type of MS [7]. MS-related fatigue (MSRF) manifests as exhaustion and a lack of energy, which is increased by physical activity and aggravated by the symptoms of MS [8]. The UK Multiple Sclerosis Society describes MSRF as “an overwhelming sense of tiredness for no apparent reason” [9]. Research has identified that it is difficult to explain and measure MSRF [10]. The cause of fatigue is poorly understood but seems to have a central primary mechanism related with a neurodegenerative process (demyelination and axonal loss in CNS) and a peripheral secondary mechanism induced by illness, complications and inactivity [11]. A recent online survey in the US explored the walking difficulties in 1011 PwMS [12]. Among the sample, 76% felt fatigue at least twice a week, 60% stated that fatigue affected their ability to participate in physical activities and 54% identified a direct impact on their employability [12]. MSRF is frequently identified as the worst symptom related to MS [13] and associated with negative feeling and psychological distress [1,11,14,15]. 

The direct cost of care for PwMS is extensive. For instance, in the financial year 2015–2016 almost 25% of PwMS had an emergency admission in hospital which cost the National Health Service in the UK £46 million [16]. During inpatient care, information provision and support around MSRF is essential. However, limited access to specialist knowledge about MSRF has been reported. For instance, in 2017, 64% of PwMS in the UK did not have regular access to a neurologist or a MS-specialist nurse [17]. 

Medical treatments seemed to be ineffective for managing progressive MS but there are some therapeutic options which appear beneficial in reducing the incidences and severity of relapses on RRMS [18]. In contrast, physical activity and exercise training (aerobic, resistance, combined training, as well as other modalities such as yoga, aquatics) have been reported to restore stamina, enhance well-being, and reduce MSRF symptoms. These benefits are important when it is also identified that there is a low risk of negative side effects [1,6,15,19,20,21]. 

A systematic review of 26 randomized control trials considered the potential risks of exercise training in PwMS with mild to moderate disability and revealed 4.6% relapse risk for the training group, compared with 6.3% for the non-exercising control group and stated that the exercise training is safe for PwMS [20]. The rate of accidents during exercise was identified as 1.2% and 2.0% for the control and exercise group respectively. The review concluded that the relative risk of adverse symptoms for the exercise group was 1.67%. This was higher than the control group but similar to the number of accidents identified during exercise training in healthy population [20]. 

Recent guidelines suggest to PwMS who have a mild to moderate level of disability participate in moderate intensity aerobic activity for 30 minutes twice per week and strength training exercises twice a week, in order to reduce MSRF levels, improve mobility and well-being [22]. However, only 20% of PwMS in the US were meeting these recommended guidelines for physical activity [23]. Several factors may prevent this including the severity of the disability, the individual education and the employment status [23]. 

A systematic review of evidence [24] has identified that exercise training can increase MSRF and highlighted that PwMS need to be supported for managing fatigue. However, within this review studies that focused on fatigue were limited and could not provide an in-depth consideration to explore the impact of MSRF [24]. A more recent review [25] that included qualitative and quantitative studies mentioned fatigue, however it did not elaborate on MSRF further than saying that fatigue represented a physical activity barrier in 10% of people. Summarily, recent review-based evidence has only been able to identify MSRF as a barrier and adverse symptom and called for further understanding of MSRF [26]. Therefore, currently researchers have been prevented from exploring and illuminating the significance of MSRF as a physical activity and exercise barrier. 

Given the above, the purpose of this review was to examine the fatigue related consequences from exercise or physical activity in PwMS. The objectives are to (a) review qualitative research to identify how and at what extent PwMS are affected by MSRF, (b) to distinguish internal barriers which keep them inactive, and (c) to reveal the negative effect of the given exercise training. Finally, the review will consider the reasons and the extent that fatigue perception inhibits the will of PwMS to participate in physical activities. A thematic analysis and synthesis [3] is a qualitative based review that was judged most suitable to gain an in-depth and detailed consideration of individuals exercises of MSRF and most able to achieve the above purpose and objectives.

## 2. Methods 

### 2.1. Synthesis Methodology

A “subtle realist” position was selected for this review. This philosophical position has been successfully used in past reviews that have considered PwMS [27]. A thematic synthesis was completed in three stages: 1) search of relevant studies; 2) critical appraisal of literature; 3) thematic synthesis. During the final stage a model was developed as a line of argument synthesis, typically used within this type of review. The review protocol was registered with PROSPERO (Ref: CRD42018115615). A pre-planned electronic search was undertaken by SM who screened all articles. 

### 2.2. Eligibility Criteria

The eligibility of each article has been considered with the use of the “SPIDER” tool [28], and articles were selected for eligibility if they matched the following criteria: 

Sample: Adults diagnosed with any type of MS. 

Phenomenon of Interest: Studies were required to report on the impact of MSRF during and after physical activity and exercise. Studies were required to consider necessary adjustments made in order to maintain physical activity in their everyday life. In addition, articles were included where health care professionals and family members of PwMS presented their own perception of MSRF.

Design: Any type of qualitative design and mixed method studies (while the qualitative component had to include at least two paragraphs of results devoted to MSRF) were included.

Evaluation: The articles were required to register and report the expressions and experiences associated with fatigue of patients or health care professionals or patients’ family members. The following methods were included: diaries, interviews, open questionnaires, surveys and focus groups.

Result Type: Studies required to include qualitative results based on observation, or interviews recording verbal and body language interaction or listed notes. There was no restriction on the date of the publication. Systematic reviews, theses, quantitative studies, and studies not written in the English language were not included in this study.

### 2.3. Electronic Search Strategy 

Five electronic databases including: MEDLINE, PubMed, CINAHL, SPORTDiscus and PsycInfo were searched from inception until 1st November 2018. Database key words included: Experience or Expressions and Fatigue and Physical Activity and/or Exercise and multiple sclerosis and Qualitative Studies or Mixed Methods. Standard Boolean operators were used as indicated above. Additional manual searches were undertaken across the first 20 pages of results from the generic web search engine Google Scholar. Additional searches of all included studies were undertaken using citation chasing and by examining the research profiles of first authors from each study. 

### 2.4. Study Screening Methods and Data Extraction

The article abstracts were blinded and sent to author AS who independently considered all selected articles for eligibility. Author SM and author AS arranged a time to discuss the inclusion of each article. Any disagreement was resolved through constructive discussion. This removed the need for a third independent academic to aid this process. SM extracted the data from the included studies using a pre-determined demographic form, where the following information was recorded: study, methodology and method (information on the interview type, duration and schedule), participants (sample size, gender, age and diagnosis), sampling (sample eligibility), setting (geographical location, place of data collection), data collection tool, aims of study, analysis type, and key themes.

### 2.5. Critical Appraisal

SM and AS blindly assessed the quality of the studies with the pre-determined and well-established critical appraisal tool (COREQ) [29] modified for reviews [3]. They also evaluated the studies to identify “fatally flawed” qualitative papers according to criteria proposed by the National Health Service National Electronic Library for Health [30]. The purpose of this was to identify papers with methodological quality that was so poor that the results became questionable.

### 2.6. Thematic Synthesis

The synthesis occurred in three stages. The data selected for synthesis included all of the results sections from each article. The initial stage involved a blind open coding line by line by SM and AS of the tabulated result sections from each included manuscript. During the second stage, descriptive themes were developed. At the third and final stage, analytical themes were generated. These were followed by various synthesis techniques in order to identify findings in a meta-analytical way [31]. The results were used to identify a model which represents an attempt to go beyond the data within thematic synthesis and is common in qualitative reviews. This is not to establish a ‘truth’ but a common process which may challenge current considerations of the topic and should be critically considered further. An audit trail for each stage of the synthesis can be considered within the Appendix A. 

## 3. Results

### 3.1. Search Output

A total of 561 articles were screened of which 19 articles were included. The PRISMA diagram describing this can be seen in Figure 1. The study designs included 18 qualitative studies and one mixed method study [32,33,34,35,36,37,38,39,40,41,42,43,44,45,46,47,48,49,50]. A total of 263 individuals were included across the studies. Two hundred forty-three PwMS (159 females, 70 males and 14 unknown) were identified. In addition to this, there were five spouses of PwMS and 15 health care professionals (six physiotherapists, six occupational therapists, three MS support workers and three neurologists). Fourteen of the studies used semi-structured interviews (14/19, 74%). The remaining studies used focus groups (2/19, 11%) and both semi-structured interviews alongside focus groups (3/19, 16%). 

Their mean age was 53.3 years and the mean time living with MS post diagnosis was 11.3 years. The diagnosis of MS type of the included participants was: 24 patients with primary progressive (PP), two patients with progressive remitting (PR), 23 patients with secondary progressive (SP), 71 patients with relapsing remitting (RR), one participant with benign (B) and 122 participants did not know their type of MS. Only four studies [32,38,40,51] identified the Extended Disability Status Scale (EDSS) scores. This meant that EDSS scores were known for just over a fifth of participants (59/263, 22.4%) and in three studies (3/18, 16.7%) [32,38,40]. In two studies [32,38] EDSS was reported as a range, between four and six for both studies (12/59, 20.3%). In another study [40] EDSS was reported in the range of 5–6.5 (14/59, 23.7%). Finally, one study with two data sets [51] reported an average of 3.8 (29/59, 49.1%) and 3.0 (4/59, 6.8%). Further demographic information of the participants can be found on Table 1 and the analytical demographic and methodology characteristics of included studies are presented on the Appendix A. 

### 3.2. Critical Appraisal of the Included Studies

All 19 studies were identified with total COREQ score that ranged between 6/13 and 9/13. No study was judged as fatally flawed, so all studies were included in the review. Between the studies, three of them had the lowest (6) and three the highest (9) score, and the most frequent score was 8 (identified in six of the papers). The findings of appraisal through the studies revealed that the first was the weakest domain with mean score 1.9/5. The critical appraisal revealed that most of the writers were not given necessary information about the identity and professional knowledge of the interviewers, consideration to bias and previous relationship with participants. The third domain had the highest mean score: 2.6/3 and the second domain had mean score 2.9/5. The summary of results of the 13 items COREQ appraisal [3] can be seen in Table 2, whereas the full table can be found on Appendix A.

### 3.3. Synthesis 

The results section is limited to the cases that were found at least 4/19 reference codes on the reviewed studies. Following that procedure, three main themes were identified, these include: (1)Fatigue-related consequences.(2)Exercise-related barriers affecting fatigue.(3)Factors that make fatigue bearable to MS individuals.

The second theme (exercise related barriers affecting fatigue) is not analysed within the results section below. This is because these results have already been extensively discussed in previous reviews [24,25,26]. However, this information is provided within the full thematic summary within the Appendix A and this information is referenced where appropriate in the discussion section. The percentage values in parenthesis refer to each sub-theme supporting studies over the total number of studies. When sub-themes were very closely associated, they were grouped together. 

### 3.4. Fatigue Related Consequences

This theme presents the most significant (a) physical, (b) emotional and (c) social consequences of fatigue which PwMS experience during and after physical activity. Analytical information about the selected type of exercise or the interventions is presented on the Appendix A.

(a) As physical consequences three sub-themes have been identified and presented below: (i) reduction of bodily control (84%); (ii) cognitive effects (32%) and (iii) increased duration of activities (21%).

(i) Reduction of bodily control 

This sub-theme is supported by sixteen studies (16/19, 84%) [32,33,34,36,37,38,39,40,41,43,44,45,46,47,48,49]. It identifies the perceived increase of fatigue during or after physical activity which causes a lack of exercise motivation and participation, either because the patients could not cope with it or by the fear of exacerbation fatigue symptoms related to excessive commitments. It includes the negative emotional response provoked by fatigue and overlapping strategies and their influence on participants’ choices to stop exercising selectively or totally. Following engagement in exercise, participants mentioned they perceived several negative fatigue-related functional outcomes. Examples included; functional decline [32,33,40], limited energy [33,39], lack of balance [33,43,48], raised body temperature [36,48,49], tripping [33], heaviness [44], knees buckle [48], pins and needles on legs [36], and reduction of reflexes in driving [48]. These outcomes were experienced as a consequence of everyday life commitments [32,33,38,39,40,46], during and post episodic or structured exercise [32,33,34,37,38,39,40,43,44,46,49] and when exercised over “the edge” (post-exertional) [40,43]. Participants identified a need to avoid the experiences of fatigue and the sense of unhealthy/overwhelming tiredness [34,37,40,43] as during such experiences they reported that they could not continue to engage in any structured activity [32,33,34,36,38,39,40,41,43,44,46,49]. 

This was despite the recognition by individuals that the symptoms of fatigue do not last long [6,37,39] and after the first difficult period, exercise might increase their energy levels. Several participants explained experiences like the following: “I will go for a walk on the treadmill... and my legs will be all jellylike, but after a few minutes—10, 15 minutes—you recover again” [46]. Individuals who did not perceive this sense of fatigue as an incomparable factor were able to monitor their fatigue levels to continue enjoying their physical activity [34,40]. However, some individuals could not potentially muster the necessary energy to overlap this difficult stage [39]. For instance, a participant stated that: “It’s overwhelming. It’s the way I’d say, you just, you don’t really care about anything but wanting to have a rest” [39]. 

(ii) Cognitive effects

This sub-theme is supported by six studies (6/19, 32%) [33,41,44,45,46,48]. It identifies the perceived cognitive symptoms related to fatigue which might affect MS individuals after physical or mental tiredness. Participants described the reduction of their cognitive capacity [33,41,44,45,46,48] which developed the following symptoms: limited memory [41,48], lack of concentration [44,48], lack of attention [48], "loss of judgment and/or perception" [48], a mental fogginess [44], “brain-cheese”, a “hazy, out-of-body fatigue feeling,” “a hangover” [46], "feel like being in a dream”, “feel sluggish” and “feel groggy” [48]. PwMS experienced these symptoms isolated or combined with the physical sense of fatigue which might be associated with emotional irritability and depression [44,45,48]. Participants described them as “different states of fatigue” [44]. Some of them experienced more often one than another, or occasionally one state of fatigue might trigger another [44]. 

(iii) Increased duration of activities 

This sub-theme is supported by four studies (4/19, 21%) [33,38,46,47]. It identifies an increase on the duration of physical activities, as a consequence of fatigue and MS impairment. This impairment was related to energy loss [46], it seemed to be worse on early morning activities [38,47] and influenced the individuals both physically and emotionally [33,38,46,47]. PwMS explained how this impairment inhibited them exercising publicly and some of them chose to exercise on their own [33].

(b) The following sub-themes have been identified as emotional: (i) negative feelings related to fatigue (53%), (ii) anxiety of pushing themselves over the limit (42%), (iii) fear of falling (26%), (iv) and anxiety for the future (26%), (v) stress (32%), and (vi) depression (32%).

(i) Negative feelings related to fatigue

This sub-theme is supported by ten studies (10/19, 53%) [33,38,39,41,43,44,45,46,47,48]. It identifies how PwMS experienced fatigue on their body and through their entire life. Participants described that fatigue caused a sense of heaviness on muscles, head and whole body [39,46], unsteadiness on their limps [47], a sense of emptiness when the energy was gone [46,48], and triggered deteriorating balance [46]. Participants’ narratives explained their individual experiences and the perceived effects of fatigue: “Wearing a trench coat that goes down to your ankles and it’s made of lead.” [46], “Your body just shuts you down” [39] and “After exercising you feel sort of emptied out and your…body’s sort of empty” [46].

The physical effects of fatigue might cause the loss or reduction of many meaningful and/or pleasurable physical activities [33,38,39,41,43,44,45,46,47,48]. In addition, the inability of some individuals to adjust their goal with the developing physical deficit [46], combined with the disengagement from highly valued activities [46] such as work, driving, exercising, social relationships [33,46,47] might resulted in a sense of permanent loss of things [39,47]. For instance, a participant articulated her experience since she gave up painting as: “A temporary sort of grieving time. It’s a loss” [47]. These participants perceived fatigue as a threat to their self-identity [46], as the hardest symptom of MS to manage [43] and believed that fatigue had complete control over them [43,46,48]. This belief was often followed by negative feelings including feelings of failure, fear, anxiety, anger, sadness, depression, and expressing helplessness over disease [39,41,43,44,46]. Unfortunately, some participants after some stressful events adopted negative coping styles, such as being in denial about MS [41] and refused to do anything or to go anywhere [38]. 

(ii) Anxiety of pushing themselves over the limit

This sub-theme is supported by eight studies (8/19, 42%) [33,34,36,39,42,43,44]. It identifies the participants’ perceived fear of over-pressing themselves over the limit in order to avoid experiencing the worsening of fatigue related symptoms which could last for several days. There was a general consensus that there is a “fine line” between beneficial and damaging exercise, so if someone crosses this line, they might experience adverse effects. The perceived consequences of the over exertion were temporary: loss of physical capacity and increased fatigue [33,36,39,43,44] affected their mood, made them feel negative, inadequate [43,44] and they had a sense of uncertainty about their ability to perform both present and future physical activities [33,43]. Participants were frustrated when some people pressed upon them to do more [44] without recognising their limits because they were afraid of the harmful consequences of over exertion [33,34,36,39,42].

(iii) Fear of falling

This sub-theme is supported by five studies (5/19, 26%) [32,36,39,43,44]. It identifies that participants who had a level of functional decline [32,43] or they have had a past negative experience of falling and/or injury, might experience a sense of fear. One cause of this was when exercising in an environment which was recognised as less safe [32,44]. For instance, a participant stated: “Going to the gym was just too hard and too treacherous. Too many opportunities to trip and fall” [32]. This negative response appeared to be associated with the belief that any potential injury would have more severe consequences for them than for a healthy person or might cause a potential exaggeration of MS symptoms [32,39,43].

(iv) Anxiety for the future. 

This sub-theme is supported by five studies (5/19, 26%) [33,39,43,44,47]. It identifies that some of the participants felt a permanent [33] or a temporary loss of control on their body. The primary causes of this included; (a) worsening MSRF [33,39,44], (b) unknown diagnosis which caused uncertainty and worry when considering the illness progression [47]. This could negatively impact their perceived ability to perform present and future physical activities [33,43]. This in turn could influence their future plans, choices and aspirations [43] and may lead individuals to think that any effort for improvement is in vein (44). For example, a participant stated: “everything goes to pot” [44].

(v) Stress and (vi) depression

Both the stress and depression sub-themes were supported by six studies; stress (6/19, 32%) [41,42,44,45,46,48] and depression (6/19, 32%) [39,40,43,44,45,46]. These studies identify a two-way association between stress and/or depression with MSRF related to physical activity. Participants stated that when they experienced an increase on stress, their fatigue perception consequently rose, and similarly depression increased fatigue symptoms [43,44] which in turn caused further fatigue. In many cases depression was presented with other fatigue consequences like: cognitive problems, lack of balance, energy, or muscle weakness [44,45]. Across the studies, participants described the distress of fatigue experience [45,48] while many of them attributed fatigue on psychological stress related to family, work or emotional problems as well which in some cases could cause a relapse [41,44,45,46,48]. A participant explained how he made the decision to stop working before he was fired and when he resigned and returned home, he had experienced a relapse [46]. Participants with a low level of physical activities when they experienced stress in their life did perceive an increased sense of fatigue; their ability to exercise was restricted and they often adopted negative coping styles [41]. In contrast, more active participants chose to exercise for reducing their stress level [41,42,44]. Furthermore, control over fatigue might cause an increase in their stress and a potential exacerbation of their symptoms [44].

Fatigue impairment provoked depression feelings [43,44,45] when PwMS: realised their inability to achieve their goals [46], felt fear for the future [44], compared themselves with healthy people [40], or when disengaged from valuable activities [39]. However, the frustration related with inability to achieve goals in some cases might be the onset of valuable adaptations [46]. Moreover, a patient described how the physical effects of fatigue depended on depression levels: “Well, it tends to … my frame of mind as well, you know if I can sink into quite deep depression as well and when I feel really low I tend to not have a lot of energy, and then other days I can be feeling really good and yeah I feel sort of happy and full of energy” [43].

(c) Social

The following sub-themes have been identified as social: (i) Imposed daily planning (37%), (ii) Dependence and the affected relationships (37%), (iii) Effects on Employment (32%), (iv) Comparison with healthy people and social isolation (21%).

(i) Imposed daily planning

This sub-theme is supported by seven studies (7/19, 37%) [33,38,39,42,44,47,49]. It identifies the necessity of planning and organizing the required activities in order to overcome the limitations of perceived fatigue. Across the studies the most frequently used fatigue-related coping strategy is planning daily [38,42,47,49] or weekly [39,49] activities. It should be noted that this strategy was identified in up to 84% of people [49]. Participants declared that everything should be planned [33,38,39,42,44,47,49], even the simplest daily activities, to have the necessary energy to complete them [42,47,49] and they also attempted to predict the duration of their time-limited energy for each desirable activity [38,39,47,49]. A significant parameter of the planning is prioritisation [39,47,49], or it may be used as imposed by the circumstances [49] and often participants should determine what is more important to do and chose some tasks over others [39,47,49].

(ii) Dependence and the affected relationships

This sub-theme is supported by seven studies (7/19, 37%) [33,38,44,45,46,48,49]. It identifies how fatigue influences MS individuals’ relationship with their partners because of the perceived physical limitations and the growing need for help from others which shattered their independency. Across the studies, participants expressed the perceived dependency for performing simple daily activities [33,38,39,44,46] and for participating in physical activities [33,46] as well. Participants also described that fatigue affected their relationships [45], increased their tendency for isolation [38], caused them to have vulnerable feelings [33] and made them feel severely limited and unable to control their body and entire life [33,38]. However, MS individuals preferred not to use mobility devices in order to maintain their independency [44,49] and physical capacity [49], as they avoided being visually different [49] to other people.

(iii) Effects on Employment

This sub-theme is supported by six studies (6/19, 32%) [41,44,45,46,47,48]. It identifies the fatigue effects of physical competence reduction on their employability [41,44,45,46,47,48], rather than MS’s functional disability [47,48]. Men seemed to be affected more from that because they became reliant on their wife’s income [46] and their self-identity was challenged [46]. Some participants replaced their demanding work with a part time job [48], while others chose a social activity for coping with work loss [41].

(iv) Comparison with healthy people and social isolation

A comparison with healthy people is supported by four studies (4/19, 21%) [40,44,45,46]. Social isolation was supported by four studies (4/19, 21%) [33,38,41,48]. These two sub-themes were inter-related. Both identify the difficulty experienced by PwMS undertaking physical activity or exercise publicly and their negative emotions when comparing themselves to healthy people, which inhibits them from participation on exercise and social activities. Participants mentioned that exercising in a public environment caused an obvious comparison between them and healthy people and considered their own behaviour not socially acceptable, so they excluded themselves from exercise [33,48]. Although these differences were not always in appearance, PwMS felt different [40,44,45,46] or disabled [44], so they were reluctant to exercise publicly [40,44,45,46]. Participants reported feeling embarrassed to explain their physical limitations to their social peers [44,46] and expressed their worries that members of the public perceived them as intoxicated because of their lack of balance [44]. The experienced restrictions on physical capacity limited them to engaged in social activities as well [33,38,41,48].

### 3.5. Exercise Related Barriers Affecting Fatigue

As explained above, this theme presents the most significant parameters related with fatigue which discourage individuals with MS to have physical activities. It includes eight sub-themes as described below:

(i) Lack of patients’ information 

This sub-theme is supported by ten studies (10/19, 53%) [32,33,35,36,38,39,41,44,45,47]. It identifies the lack of patients’ knowledge about the exercise benefits and how it might minimise their fatigue symptoms. Negative beliefs related to the efficiency of exercising prevented the MS individuals from making any effort to continue being active.

(ii) Lack of motivation and support 

This sub-theme is supported by ten studies (10/19, 53%) [33,35,36,38,40,41,42,44,45,46]. It identifies the perceived need for motivation and support by health care professionals, family and friends. Often the motivation for providing support was to aid the patient’s physical limitations. In some cases, the provision of social support by family and friends’ behaviour acted as a barrier for them to be active.

(iii) Lack of appropriate knowledge/ understanding of exercise providers 

This sub-theme is supported by nine studies (9/19,47%) [33,35,36,38,40,42,44,45,50]. It identifies the lack of knowledge of health care professionals and exercise instructors and also the patients perceived gap in optimal communication between professionals and patients. This inadequate information provision and advice may have contributed to a belief by some PwMS that they, rather than the health care professional knew what was suitable for them. This often resulted in a choice to undertake exercise independently or stop exercising.

(iv) Geographical distance and financial difficulties 

This sub-theme was supported by seven studies (7/19, 37%) [32,39,40,41,45,47,49]. It identifies the perceived barriers on MS individuals to access athletic facilities. Geographical distance, lack of financial support, lack of disabled parking and difficulties to use various means of transportation are some of the mentioned barriers which finally prevented some from partaking in regular exercise.

(v) Conflicting recommendations 

This sub-theme is supported by four studies (4/19, 21%) [36,38,39,45]. It identifies the ambiguities of health professional advice regarding exercise and its effects on patients’ motivation. Health care professionals described how physiotherapists and occupational therapists had a different approach with regard to fatigue management and this “interprofessional conflict” had negative influence on the patients. In addition, the complex nature of MSRF appeared to create confusion over the ownership and scope of roles require from each professional group. For instance, one study stated that “some participants perceived their roles to be undervalued and, at times, poorly understood or undermined by other members of the healthcare team” [45].

(vi) Busyness or other interests preventing exercise 

This sub-theme is supported by nine studies (9/19, 47%) [32,33,38,39,41,42,44,45,47]. It identifies the perceived personal commitments which competed exercise. Physical activity could be limited in participants who prioritised other social role such as caring, paid or voluntary employment. The main consequence of this was limited energy reserves were available to accomplish physical activities or exercise.

(vii) Type of exercise 

This sub-theme is supported by nine studies (9/19, 47%) [33,34,35,39,40,41,44,46,49]. It identifies the perceived individual difficulties from specific types of exercises with the participants’ corresponding symptoms. Participants with physical deterioration avoided vigorous exercise such as aqua aerobics, chair-based yoga [32] or classic yoga [33], running [33], normal road bike [33], and walking outdoors [44]. The main reason for this was because of the fear of falling. There was a preference for specific types of exercise equipment. This included the stationary cycle [44] and treadmill [44]. Both types of equipment which gave them a sense of safety. Other reasons included the lack of environmental barriers, e.g., “there’s no paving stone to trip over” [44]. Across the studies participants avoided [33,40,41] or chose [34,44,46] to participate in the same activities for just the opposite reasons. For example, participants with poor balance reported feeling unsupported and anxious to exercise in swimming pools [33] while another stated that when exercised in water gained a sense of normality [44]. Some avoided swimming pools because they were warm [41], while others chose swimming for staying cool [46]. Swimming increased the energy levels in PwMS [39]. Importantly, exacerbation of MSRF with swimming was reported in two studies (2/19, 10%) [34,39]. Participants mentioned the significance of monitoring their fatigue levels to be able to continue and enjoy their selected activity [34]. Endurance exercise was often perceived to increase body temperature, so often was avoided by individuals as this increases MSRF [49]. 

(viii) Weather 

This sub-theme was supported by six studies (6/19, 32%) [32,41,44,46,48,49]. It identifies the perceived weather-related barriers and their consequences on the symptoms of PwMS. This often caused a temporary discontinuation of weather dependent activities. Across the studies many participants reported exaggeration of fatigue-related symptoms and lack of energy in hot weather [32,41,44,46,48]. It has been stated that humidity influenced the energy levels and made the muscles stiffer [48]. In cold climates, low temperature caused stiffness and pain, affected balance and might influence the energy levels [44,46,48], while wind inhibited some participants from walking outside because they felt unsafe [44]. Most of the participants expressed their difficulty to continue exercising in extreme outdoors temperature [32,41,44,46].

### 3.6. Factors that Make Fatigue Bearable to MS Individuals

This theme presents the most common motivator factors which support the willing of MS individuals to remain active.

It includes eight sub-themes: (a) appropriate professional guidance (42%), group exercise (53%) and social support (37%), (b) exercise benefits (68%), (c) exercise experience (58%), (d) appropriate level &/or type (47%) and enjoyable exercise (21%), (e) belief that exercise increases energy levels (53%), (f) patients’ education (32%) for self-management and control over fatigue (32%), (g) acceptance of MS and determination (42%) for priority shifting (37%), (h) adaptability and positive thinking (47%), (i) rest, pleasant activities (26%), and cooling strategies (26%).

(a) Appropriate professional guidance, Group exercise and Social support

This sub-theme is supported by thirteen studies in total. This included eight studies (8/19, 42%) [32,35,36,40,44,45,47,49] for appropriate professional guidance, ten studies (10/19, 53%) [32,35,36,37,38,41,44,46,47,50] for group exercise and seven studies (7/19, 37%) [38,41,42,44,45,47,50] for social support. These sub-themes identify the perceived needs of PwMS regarding exercise support and the benefits from appropriate professional guidance and group training. It includes the value of family encouragement and social support in order for the PwMS to remain active. Across the studies participants highlighted the significance of person-centred professional guidance from an exercise specialist trained for a MS population [32,35,36,40,49] who offered them feedback and advice [44], was able to understand individual capabilities and limits [33,35,44,49], and suggested appropriate exercises adapted to MS variability [47]. Participants reported the perceived benefits from this formal guidance as they felt safe, supported [40,44] and confident [35], remained engaged in exercise [35,44], enjoyed it [32,40], memorised helpful tips [44], learned how to pace themselves [36], and recognise their limits [35,44].

Many participants highlighted the value of group exercise to developing a sense of motivation and support [35,38,46,50] and realised it was easier to exercise in a group than at home [35,38,50]. Participants emphasised the importance of exercising with individuals who had similar difficulties [37,38,40,41,47,50], as they were able to improve learning and gain experiences by sharing their mutual problems and through the interaction amongst the group [38,50] which provided them with encouragement and inspiration to try harder [37,38,40]. Participants felt: safe [37], confident [37], encouraged [37,38], empowered [40,50], their attitudes improved [38], they stopped feeling sorry for themselves [40], felt normal [44] and accepted by others [44]. 

(b) Exercise Benefits

This sub-theme is supported by thirteen studies (13/19, 68%) [32,34,35,36,37,38,40,42,43,44,47,48,50]. It identifies the perceived physical, functional and psychological benefits following exercise and the positive consequences on participants’ quality of life, their way of thinking and their ability to undertake physical activities. Individuals highlighted the direct benefits of exercise on fatigue, including reduced levels of fatigue [35,37,44,48], increased endurance [37,43,44], and increased energy [35,40]. This was also described as a “healthy tiredness” [43,44]. For instance, following the exercise intervention a participant stated: “the fatigue sort of goes into the background”. [43]. Exercise was identified as a bridge to overcoming barriers to activity [40] and it enabled a sense of achievement [43]. Across studies, participants identified perceived improvements from undertaking exercise, including: engaging in new or previous recreation activities, feeling independent [32,35,36,38,43,48,50], regaining or improving strength [37,38,44,47], activity level [32,34], health [34,36], balance [38,40] and flexibility [34,37], movement getting around, and the achievement of functional tasks [34,38,42].

In addition, participants described improvement in psychological well-being as often accompanied by short statements summarised as: the intervention making them feeling better [37,38,42,47], providing enjoyment [36,37], happiness [37,38], a sense of reward [36,37,38,40,42,43,44,47], confidence towards [32,35,40,42,43] and a more positive outlook or perception of well-being [35,38,40,43,47] and ability to engage in activities. Other noticeable psychological benefits were improvement on: stress management [42,43], empowerment [35,42], and alertness [40,44]. 

Some individuals felt the exercise engagement, as an “opportunity to challenge their self-limiting thoughts” [40], enabled them to take control over their symptoms [43,44,47] and encouraged their decision to continue exercising following the end of the intervention [35,40,42]. The perceived physical and psychological improvements provided them with a sense of achievement [35,40,43,44,47] which was a significant factor in the process of coping with MS in order to “maintain their identity” [47]. 

(c) Exercise experience 

This sub-theme is supported by eleven studies (11/19, 58%) [32,33,34,35,36,37,38,42,43,46,47]. It identifies the perceived benefits by exercise experience. Participants reported that when they increased their physical capacity, they experienced a sense of control over MSRF, gained a sense of normality and were able to be more engaged in social activities. For instance, a participant stated: “The more exercise you do, the more you want to do” [32]. Participants felt as normal as before MS onset [32,37,47] and reported the development of a sense of control [32,33,43,46], accomplishment [32,36,38,42,46], and well-being, improved mood and happiness [32,36,37,38], reduced stress level [42] and felt proud of themselves [32], even though they needed to perform some adaptations in order to complete an activity [38,46]. 

Participants who had the experience of exercise effects recognised that they should underestimate the initial fatigue symptoms in order to receive exercise benefits. For instance, a participant stated: 

“You are scared because you immediately get the symptoms from the increased body temperature and everything anyway, my feet automatically have pins and needles all up my legs and that is murder and it is a sign that I will have to stop, and in actual fact what I have learnt is that it will fade, that is alright, it is your body just reacting and increasing temperature and is perfectly normal and carry on” [36].

(d) Appropriate level and/or type and enjoyable physical activity

This sub-theme is supported by eleven studies. This includes nine studies (9/19, 47%) [32,33,40,41,42,43,44,46,50] focusing on the appropriate level and/or type of exercise and four studies (4/19, 21%) [32,36,40,45] focusing on the enjoyment of physical activity. These themes identify the significance of making necessary modifications on exercise’s type and characteristics for it to be enjoyable and so enable the patients to maintain a suitable level of physical activity. Across the studies, participants described their need to scale back the intensity, duration and frequency of exercise [32,40,41,42,43,46] and also to choose the most suitable exercise from a variety of options, which minimised their physical symptoms [33,40,41,44,46]. Many participants maintained their engagement with physical activities through incidental exercise such as: walking dog [32,46], walking by the beach [44], gardening, playing with children [32], or choose a gentle exercise like treadmill [33], static bike [32,44], yoga [32,44,46,50], Pilates [40,45], or aqua jogging [44] and swimming [32]. Health care professionals highlighted the significance of enjoyment during participating in exercise activities as a facilitator for the PwMS to remain motivated [45]. Exercise has an imperative characteristic on PwMS who therefore have to find an enjoyable type in order to balance the perceived drawbacks and benefits [44]. A participant described the significance of feeling normal when participated in aqua jogging and that defined her activity choice: “I think in the water you feel like a human being again. You feel like ’you’re normal, whereas on land you ’don’t feel normal. I think ’it’s the feeling as ’you’re on an equal footing with everyone” [44]. Equally important was the choice of exercise settings, mostly for the participants who experienced high levels of fatigue and lack of balance [33,41,42,44,46]. A broadly used choice was exercising at home with gym equipment [41,44,46], where they experienced higher level of convenience, privacy and security with lower energy expenditure [44,46] that also enabled them to disperse the exercise over the day or to stop when they felt tired [41]. In contrast, participants who had strong control over their limits by “listening to their bodies”, wanted to further challenge themselves and progress on different types or levels of exercise because they felt boredom and frustration when they were unavailable to push themselves to their real limits [40,43].

(e) Belief that exercise increase energy levels

This sub-theme is supported by ten studies (10/19, 53%) [32,34,39,41,42,43,44,45,46,48]. It identifies the fact that exercise increases the energy levels and decreases the perceived fatigue. Across the studies participants reported that taking part in physical activity created energy [32,39,42,44,48], made them feel better [39,44], improved long term fitness [39,43,44], increased strength and endurance [44], gave a sense of “healthy tiredness” [34,44], increased the control over fatigue [41,42,44,48], and led to a greater participation in physical activity [39,43,44,45]. Participants reported that even though sometimes they initially experienced a rise in fatigue, they felt better when they continued exercising [39,44,46]. Moreover, when they felt tired, they needed to exercise in order to experience an increase in their energy levels [32,39,42]. Conversely, if participants remained inactive, they had experienced lower energy levels and physical deconditioning [39,48].

(f) Patients’ education for self-management and control over fatigue

This sub-theme is supported by ten studies. This included six studies (6/19, 32%) [36,39,40,42,44,50] identifying patients’ education and six studies (6/19, 32%) [41,43,44,45,46,50] identifying perceived control over fatigue. These themes were inter-related and identify the perceived benefits from education about exercise benefits and fatigue management. Participants reported that learning about fatigue management [39,50] and exercise effects [36,39] enabled them to explore the mystery about exercise, felt stronger, confident [36], and altered their previous thinking and guilty emotions around fatigue [50]. However, even though some participants had an adequate knowledge about exercise benefits, they “avoided pushing themselves beyond their limits” [42] because such knowledge alone could not result in a higher sense of responsibility for managing their symptoms [41,50]. Knowledge combined with exercise experience empowered them to determine their engagement in physical activity to increase control over fatigue [41,45,50] and to achieve a self-management. MS individuals adopted individualised problem-solving techniques [41,43,44,46] which enabled them to remain active [41]. Their strategies were based on experimentation by listening to “their bodies” [52,53] and were developed according to their disability level and beliefs. Some chose safe strategies with small gradual steps to avoid overdoing and turning back [44], the more confident exercised beyond the “edge” by controlling the perceived fatigue with rest-periods and rehydration [43]. One of them observed that: “complete rest after these exhausting periods was not helpful, and if he could motivate himself to go for a swim or do some exercise, he could get through that feeling of fatigue” [46]. Participants who achieved a high level of self-management, reported a strong sense of control over fatigue [41,43,45,46,50], were able to push themselves to their limits [43], adapted their energy requirements every moment [43], had self-efficacy, body awareness [45] and better perceived exercise outcomes [43].

(g) Acceptance of MS and determination for priority shifting

This sub-theme is supported by eight studies (8/19, 42%) [33,38,39,41,42,44,45,46]. It identifies the importance of PwMS determination to remain physically active, even though they fight with fatigue related symptoms and needed to modify their goals in order to be able to exercise regularly. Patients who accepted the presence of MS in their life could be determined to continue exercising despite the difficulties and that empowers them to overcome fatigue [42,45]. Participants that were conscious about the consequences of not exercising [41] and decided not to focus on the limitations caused by MS were able to make the appropriate adaptations to optimize their health and accomplish the highest possible standard for their life [41,42,45].

Participants prioritised exercise, as a part of their daily routine [42,45] because they felt better [42] and believed that exercise on daily base empowered them to maintain their functional improvements [45]. Some participants decided to buy a dog in order to make a commitment to walk every day [32,41]. Others reported that they carefully scheduled the type and intensity of exercise [42] and integrated it into their daily commitments [39] in order to enjoy exercising without overexerting themselves [39,42,46].

(h) Adaptability and positive thinking

This sub-theme is supported by nine studies (9/19, 47%) [36,40,41,42,43,46,47,48,49]. It identifies the perceived improvement. This includes a development of positive feelings and control over MSRF, by planning ahead for pre-empting situations that may cause fatigue and making adaptations on meaningful physical activities to prevent too greater impact of MSRF. Across the studies, participants reported many different strategies which empowered them to reduce fatigue experience and remain active [36,40,41,42,43,46,48,49]. This contributed to a sense of optimism [40,41,42,43,46,47] and to participate in social roles [41,47]. Participants reported their attempt to “work smart” in order to maximise performance and minimise the energy cost [48]. However, some of these strategies imposed readjustments on MS individuals’ standards and expectations [49]. These decisions might include the modification of exercise characteristics, but also the perceived value of an activity in order to achieve a meaningful goal [36,43,46]. 

(i) Rest, pleasant activities & cooling strategies

This sub-theme is supported by nine inter-related studies (9/19, 47%) [32,41,42,43,44,46,48,49]. This includes five studies (5/19, 27%) [32,42,43,46,49] that consider rest and pleasant activities and five studies (5/19, 27%) [41,44,46,48,49] that consider cooling strategies. These themes collective identify the perceived benefits of some useful strategies to manage fatigue effects during exercising. Participants were planning their activities such that they were having rests between them, or using them when necessary [32,42,43,46,49]. Participants reported the use of short breaks, with or without a cup of tea, as beneficial, in order to recover within an activity for reduction of fatigue symptoms [32,43,46]. Another strategy used for restoring physical energy was to undertake diversional or pleasant activities to distract their attention from the distress of experienced fatigue [48]. PwMS also depicted their need to implement a variety of cooling strategies [41,44,46,48,49] to reduce the negative influence of hot weather on perceived fatigue, during exercise [46].

### 3.7. The model of physical activity facilitation and inhibition on MSRF

Figure 2 illustrates the model of physical activity facilitation and inhibition on MSRF. This model is illustrated by a two-cycle process. The outer cycle illustrates a six-stage process of enabling physical activity. This cycle captures a process of positive management of MRSF and exercise facilitation. The inner cycle identifies how individuals may succumb to the effects or experiences of MSRF. When different domains of the inner cycle are experienced, this may develop into a vicious cycle of inactivity. 

The six stages of the outer cycle can be considered within domains as follows: Stage 1 identifies the positive perceived past experiences of the exercise. The positive experiences represent a starting point for engagement in physical activity and are likely supported by positive past social support and environment during the activity. Stage 2 identifies that an appropriate level and type of physical activity enables PwMS to maintain engagement. Stage 3 identifies that determination to hold an active coping strategy empower PwMS to remain active. Stage 4 identifies that the knowledge of physical activity benefits is a strong motivator for exercising. Stage 5 identifies that perceived control over MSRF encourage PwMS to increase physical activity. Stage 6 identifies a perception that undertaking physical activity in a successful way increases energy levels in PwMS.

The inner cycle is informed and created by negative influences from the outer cycle. Each domain of the cycle has a bi-directional link to other parts within it. The cycle may be initiated by any single stage. For instance, poor choice or a lack of knowledge about beneficial type, intensity or time committed to physical activities may cause physical decline. This in turn may create negative feelings and result in further experiences of fatigue as well as physical decline, amotivation and social isolation. Alternatively, MSRF may lead to inactivity and inhibit participation in social activities, so elicits negative feelings and further physical decline. 

## 4. Discussion

This systematic review synthesise evidence about the consequences of MSRF related to physical activity. MSRF has enormous effects on different domains including physical, emotional and social well-being. When one domain is affected our results suggest that this may trigger another domain to be affected and could fuel a vicious cycle. Kayes et al. [39] have previously identified a cycle of (in)activity and beliefs relating to benefits on energy levels. They have also identified the importance of different domains within the current model as important to the physical activity engagement. The current research furthers their work by identifying work by identifying two cycles of physical activity enablement and inhibition cycle includes more than one component, the results suggest that MSRF causes reduction of bodily control and cognitive dysfunction, which in turn provoke emotional distress, leading patients to social disruption and physical inactivity.

### 4.1. Physical activity Inhibition

Physical disability related with MSRF increases the level of stress in PwMS. A recent longitudinal study [18] considered the psychological and cognitive functions in PwMS and showed that fatigue was conceptualised as stress. The authors stated that as the ability to participate in physical and social activities was limited greater levels of stress were perceived. Depression is a common symptom in PwMS as well [17], MSRF is highly associated with depression, while disability and depression correspond with higher levels of fatigue in PwMS [1,54]. In this review it was found that when PwMS experienced stress and/or depression, experiences of MSRF was consequently increased and vice versa [43,44]. Family or work-related stress might increase MSRF symptoms and in exceptional conditions might provoke a relapse, while depression may conversely affect energy levels. High levels of stress inhibit the engagement of physical activities because of the fatigue increment. It is possible that less active individuals adopt emotion focused coping strategies more frequently. Such strategies likely compound the impact of stress. Furthermore, physical deterioration related to MSRF may increase depression when PwMS are disengaged from meaningful activities, or when they are unable to meet rehabilitation goals, or through social comparison. In this review depression appeared to be positively associated with cognitive problems and physical deterioration. Studies have shown that a sustained sense of fatigue and emotional stress can cause biochemical and structural changes in PwMS’s brain that are related to cognitive problems [55]. This may cause depression when the individuals adopt avoidance strategies [52].

The findings of this review, in agreement with previous findings, synthesise the complete cycle of exercise inhibition (inner cycle see Figure 2). Fatigue, physical deterioration and negative feelings likely have a two-way association and may cause social disengagement and inactivity. Exercise disengagement increases symptoms of MSRF and provoke physical deterioration. In addition, social isolation, apparent in PwMS can increase stress and /or depression (negative feelings) which aggravates MSRF and can inhibit physical activity. In this vicious cycle less active PwMS may not be able to overcome fatigue symptoms. These individuals may also feel anxiety due to worsening disability, which, may combine with beliefs about potential damage from exercise. At an extreme, such views may impact on the choice to participate in regular physical activity.

### 4.2. Physical Activity Facilitation

Studies revealed that task specific exercise increases neural-plasticity [56,57] and promotes synaptic and axonal growth, angiogenesis and neurogenesis [25]. Thus, exercise is not only beneficial for psychosocial well-being but also for brain health. However, recent evidence supported that the MS pathology provides a chronic cell stress and imbalance, resulting in a continuing axonal and neuronal loss [22]. This may explain the further increase of MSRF post-physical activity, which introduces supplementary stress to an already stressed nervous system. [6]. In this review there is evidence supporting that control over fatigue might cause an increase of stress on PwMS and a potential exacerbation of their symptoms. A cross-sectional study compared the coping strategies of PwMS to those of healthy people and revealed that PwMS had less ability to cope with simple everyday tasks and they also had lower mental and emotional capacity to adapt and confront their stress and other daily problems, than the general population [53].

Our findings revealed that it is mostly less active PwMS who have a high level of difficulty to confront stress related to exercise. It is possible that avoidance strategies or emotion focused coping is more prevalence for less active PwMS. Although there is a lot of evidence supporting the benefits of exercise, PwMS can have an ambiguity or anxiety when considering the dose and other characteristics of the exercise. Appropriate support by qualified professionals, communicating and sharing mutual problems with other PwMS, together with their own individualised experience from exercise may constitute an essential first step to transition towards a more physically active state. Experience of physical activity benefits is a strong facilitator for participating, but it should be achieved at the lowest cost, i.e. the lowest negative consequences. PwMS need safe conditions with low level of stress in order to reduce fatigue related to exercise. Health care professionals as well as physical activity peers can transfer their experiences to PwMS, thus providing a sense of safety and reasonable explanations about their symptoms. Social and financial support [24,25,26,36] are helpful in developing ideal conditions and reduce the perceived fatigue related with transportation and physical preparation for the exercise, but on their own are not adequate to facilitate physical activity, as in many cases inactive PwMS already had them.

The second important step is the participation in a suitable, individualised and enjoyable exercise training. All these components have been analysed in the results section and compose the necessary adaptation for creating conditions for physical activity which minimises the perceived physical fatigue post-participation and stress associated with participation. Our findings revealed a great variety of personal barriers and individual needs that make it hard to find ‘ideal’ physical activities for any one individual. Misbeliefs, previous negative experiences, and specific physical impairments shape the individual impediments of PwMS. Health care professionals and trainers may achieve greater outcomes through listening and providing PwMS choice of which activity to undertake [51].

Recent studies have identified that even when a high level of fatigue during physical activity is experienced benefits can be identified. For example, they may face less stress and cognitive problems [18] and they are protected against depression [52] if they adopted an active coping style. PwMS seem to use different coping strategies in early and later stages of the disease, depending on their gender and disability level. There is evidence that in the middle stage of the disease PwMS mainly use active and adaptive strategies and less frequently avoidance and maladaptive coping patterns [52]. This suggest that PwMS adopt more effective coping styles over time, potentially because at this stage they have already accepted the presence of the MS in their life and have less stress and anxiety about the risk and future consequences of exercise. 

It is important to note the covariates of physical activity for PwMS. For instance, a secondary analysis of previous data showed that PwMS participated less frequently in regular exercise when: they had a progressive type of MS, they lived with MS for longer period, they could not walk fully independently, were unemployed, and had a low level of education [23]. The acceptance of MS is a significant factor which enables PwMS to realise that they are responsible for their condition and facilitates the activation of self-management [51,58]. Most of PwMS are aware about the consequences of not exercising and seek to exercise for reducing fatigue and emotional stress. Many family or work-related distractions or even experience a period of aggravation of their MS symptoms, or other health problems, can inhibit the ability to engage regularly in physical activities. The physical and psychological benefits of exercise appear to increase resilience, aid coping and give individuals the know-how to modify their activities in the most suitable manner to them.

PwMS who have experienced the benefits of physical activity and likely feel a sense of accomplishment and normality. The physical and emotional effects of physical activity empower them to increase their participation in physical and social activities in order to increase control over fatigue and their self-esteem. Studies have shown that the higher self-esteem PwMS have, the more problem-focused they become and the less emotional strategies they adopt [59]. The current results suggest that less active PwMS are empowered by positive physical activity experiences. Such experiences likely result in positive changes on their behaviour, better mood and social life. It is also likely that more active individuals enjoy exploring their limits in order to optimise their physical capacity and maintain the highest possible standard of physical activity.

### 4.3. Limitations

There are a few parameters which have not been considered in our verbatim statement. For instance, resilience which is the human capacity to adapt to changes, to overcome barriers, to cope with difficulties and recover after stress, seems to be one of the foundational factors for a healthy aging in PwMS [58]. Young and middle-aged PwMS have especially low resilience and in a recent study it was stated that physical fatigue was one of the barriers of PwMS to achieve resilience [60]. In addition, to author’s best knowledge, there are no studies evaluating separately the experience of fatigue related to exercise in people with progressive or relapsing remitting MS. There is evidence that people with progressive MS adopt a more negative attitude to coping with the disease than individuals in remission stage [61] and this is unsurprising because the former experience more often the disease symptoms than the latter, who can experience periods of their life without symptoms. Some of the investigated studies also included participants with a very broad range of years post MS diagnosis, e.g., between 1 and 30 years [44]. The findings of a recent review showed that PwMS adopted different copping strategies during the different disease stages. PwMS at the middle stage of the disease used more often active and adaptive strategies than avoidance and maladaptive coping strategies [52]. This may give a broader information about PwMS’s needs but does not facilitate the introduction of specific tips for each category. The heterogeneity of data from included studies meant that the impact of self-selected samples (possibly more active), types of physical activity or exercise and EDSS scores were difficult to determine. For instance, the editors of one of the investigated studies attempted to classify the activity level of the participants post-interview, but chose to base the classification on five pre-determined questions rather than use a broadly accepted protocol [41]. In addition, there is a cross-over of the terms “exercise” and “physical activity” which are used as synonyms through the studies, because MS and MSRF may limit the ability of PwMS to participate in more structure exercise. A participant reported that “…while exercises are separate, they are linked to everyday life activities. Therefore, physical activity and exercise whilst considered different in definition based on usual types of movement, they were nevertheless on a spectrum” [47]. Further to this, some of the studies had small number of participants. Moreover, themes that have been recently analysed by other authors are not mentioned in the synthesis. This information is available in Appendix A.

### 4.4. Further Research

There is evidence that PwMS adopt different strategies to cope with fatigue as their illness progresses. It would be beneficial if future studies separately evaluated fatigue experiences related to exercise at different stages of the disease in order to reveal their experiences associated with periods in which they adopt less active strategies.

## 5. Conclusions

The model of exercise facilitation and inhibition, as presented above, integrates MS patients’ experience from fatigue consequences related to exercise. Strong evidence supported that exercise reduces fatigue to PwMS. The implementation of a suitable individualised exercise programme with appropriate social support minimises the stress of physical activity. Exercising in less stressful conditions with a suitable and enjoyable physical activity will likely minimise MSRF. In addition, exercise experience helps individuals to recognise and interpret the initial symptoms of fatigue, reduces the anxiety related to the potential negative effects of exercise, proves the exercise benefits and empowers PwMS to adopt an active coping strategy in order to break the vicious cycle of inactivity and maintain their normality.

## Figures and Tables

**Figure 1 behavsci-09-00070-f001:**
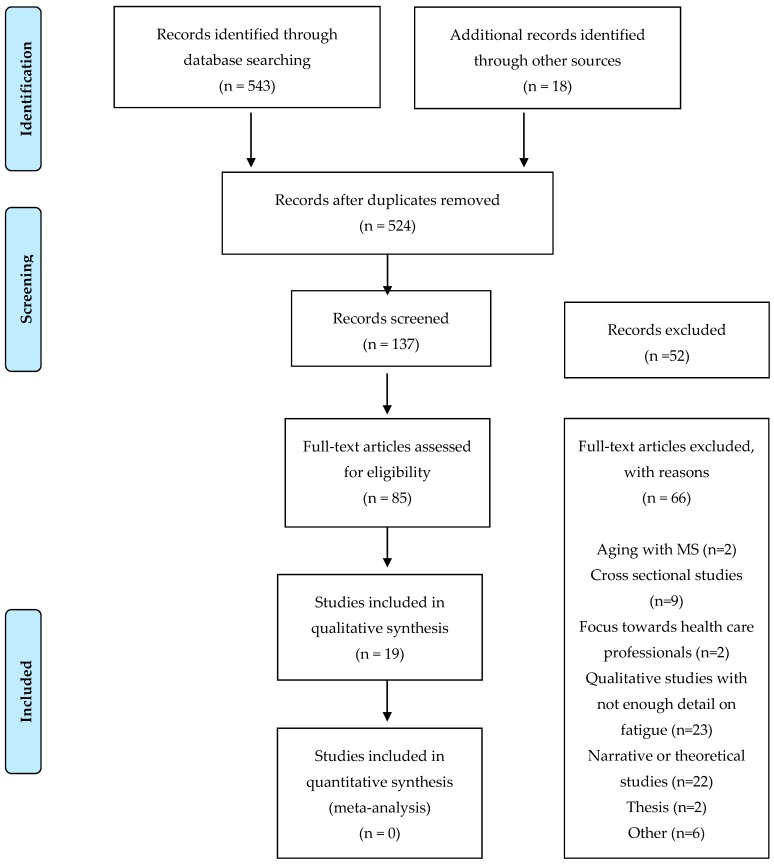
PRISMA diagram.

**Figure 2 behavsci-09-00070-f002:**
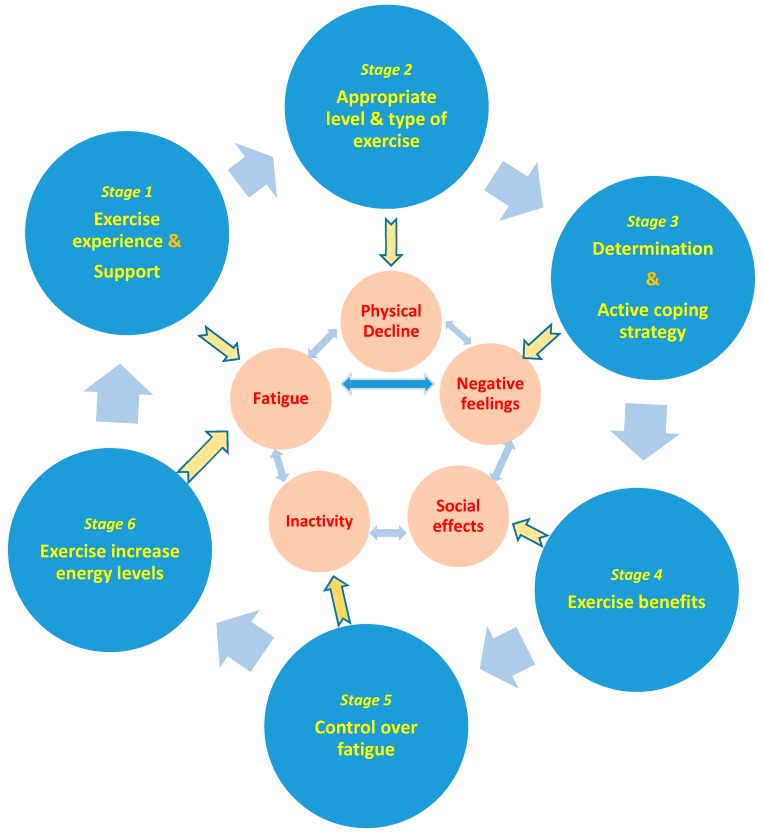
The model of exercise facilitation and inhibition.

**Table 1 behavsci-09-00070-t001:** The Demographic Information of studies’ participants.

Stakeholder	Author/year of Publication	Female	Male	Both	Type of MS	Mean Age (years)	Age Range (years)	Mean Years Since Diagnosis	Range of Years Since Diagnosis
PwMS	Barnard, 2018	13	5		Mild 11/18, Moderate 3/18, Severe 4/18	50.3		9.39	
	Borkoles, 2007	4	3		EDSS 4-6	47.1		16.3	
	Brown, 2012	5	3		4PP, 2PR, 2RR		47-66	16.9	
	Clarke, 2015			14	1RR, 2SP, 10 not known	53.9		10.3	
	Crank, 2017 1st group	23	6		EDSS 3.8	48.8		8.8	
	Crank, 2017 2nd group	3	1		EDSS 3.0	48.8		9.7	
	Dodd, 2006	7	2		Mild to moderate disability	45.6		6	
	Horton, 2015	1	4		EDSS 4-6	57.4		13.4	
	Kayes, 2011	7	3		CP 3, SP: 3, RR: 4	44.3		8.8	
	Learmonth, 2012	10	4		EDSS 5-6.5	59.6		16.9	
	Plow, 2016	11	2		SP: 2, RR: 9, Unknown: 2	46.7		12.2	
	Schneider, 2018	7			PP: 3, RR: 4	50.4		5	
	Smith, 2009	8	2		RR	46.4		13.1	
	Smith, 2011	9			SP: 3, RR: 5, 1 unknown		28-70		1-30
	Smith, 2015		18		SP: 3, RR: 10, & PP: 5		36-68		3-21
	Stennett, 2018	12	4		SP: 4, RR: 2, PP: 5 & Unknown:5	61.3	47-72		
	Stuifbergen, 1997	10	3		Not given	44.6		13.9	
	Turpin, 2015	23	8		RR: 21, SP: 4, PP: 4, Unknown: 2	51.3		10.3	
	Twomey, 2009	6	2		RR: 3, SP: 2, Unknown: 2, Benign:1	42.8		10.1	
**Total**		**159**	**70**	**14**	**243**	**53.3**		**11.3**	
Others	Horton, 2015				Spouses; 6	56.8			
	Smith, 2013	Physiotherapists; 6, Occupational Therapists; 3, Multiple Sclerosis Social Support Workers; 3, Neurologists; 3. Total; 15. [No further details given]							
**Total**		**20**							

Note: EDSS = Extended Disability Status Scale; PP = Primary Progressive Multiple Sclerosis, RR = Relapsing Remitting Multiple Sclerosis, SP = Secondary Progressive Multiple Sclerosis, CP=Chronic Progressive; PwMS = People with Multiple Sclerosis.

**Table 2 behavsci-09-00070-t002:** The summary scores of the 13 items COREQ appraisal [3].

Author/year of Publication	Domain 1: 5/5 Research Team and Reflexivity	Domain 2: 5/5 Study Design	Domain 3: 3/3 Analysis & Findings	Grand Total
Barnard, 2018	4/5	2/5	3/3	9/13
Borkoles, 2007	1/5	4/5	1/3	6/13
Brown, 2012	1/5	2/5	3/3	6/13
Clarke, 2015	2/5	3/5	2/3	7/13
Crank, 2017	1/5	4/5	3/3	8/13
Dodd, 2006	1/5	3/5	3/3	7/13
Horton, 2015	4/5	0/5	3/3	7/13
Kayes, 2011	1/5	4/5	3/3	8/13
Learmonth, 2012	2/5	2/5	3/3	7/13
Plow, 2016	1/5	4/5	2/3	7/13
Schneider, 2018	3/5	2/5	3/3	8/13
Smith, 2009	2/5	3/5	3/3	8/13
Smith, 2011	1/5	5/5	3/3	9/13
Smith, 2013	1/5	5/5	3/3	9/13
Smith, 2015	1/5	2/5	3/3	6/13
Stennett, 2018	3/5	3/5	2/3	8/13
Stuifbergen, 1997	0/5	3/5	3/3	6/13
Turpin, 2015	3/5	3/5	2/3	8/13
Twomey, 2009	4/5	2/5	2/3	8/13
**Mean score**	**1.9/5 =0.3**	**2.9/5 =0.6**	**2.6/3= 0.9**	**7.5/13 = 0.6**

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
