# Peer review of "A Thematic Synthesis Considering the Factors which Influence Multiple Sclerosis Related Fatigue during Physical Activity"

_behavsci, 2019, doi:10.3390/bs9070070_

Round 1

Reviewer 1 Report

The Review Article incorporates all specific issues concerning MS related fatigue in association with exercise. It is a very well written paper.

I have only a few questions regarding the type of exercise, which isn't mentioned much in this article, an EDSS as a limitation factor as well. The type of exercise has an influence on fatigue or can be a limitation factor for exercising itself and enjoying (i.e., for PwMS having EDSS ≥ 7). The Authors did not mentioned about the EDSS as a limitation factor for exercising in the part “Exercise Related Barriers Affecting Fatigue”, which makes fatigue even more pronounced in those people having limitation for exercising (i.e., wheelchairs). In addition, which range of EDSS they encountered in all these references they studied. Could the authors mention it? If not, please mention it in the Limitations part.

In the manuscript part „Type of exercise” (Page12 Line423-429) the authors wrote only about the exercise in water as an example. Could the authors incorporate more examples in this part beside the water as an issue for exercising (i.e., treadmill, bicycle, aerobic, endurance, running…).

Technical issues:

Page2 Line72: please explain A&E

References: the authors wrote double references (under numbers 8 and 9: Braley TJ,2010; and under numbers 36 and 57: Crank H,2017). Please check the other as well.

Author Response

Author response: Thank you for these comments.

"I have only a few questions regarding the type of exercise, which isn't mentioned much in this article, an EDSS as a limitation factor as well. The type of exercise has an influence on fatigue or can be a limitation factor for exercising itself and enjoying (i.e., for PwMS having EDSS ≥ 7). The Authors did not mention about the EDSS as a limitation factor for exercising in the part “Exercise Related Barriers Affecting Fatigue”, which makes fatigue even more pronounced in those people having limitation for exercising (i.e., wheelchairs). In addition, which range of EDSS they encountered in all these references they studied. Could the authors mention it? If not, please mention it in the Limitations part."

Author response: Thank you for this observation. We have added a summary of the EDSS scores into the results section (see Table 1). We have added details in the supplementary file for the theme as identified above. We have also added this point into the limitations.

"In the manuscript part „Type of exercise” (Page12 Line423-429) the authors wrote only about the exercise in water as an example. Could the authors incorporate more examples in this part beside the water as an issue for exercising (i.e., treadmill, bicycle, aerobic, endurance, running...)".

Author response: Thank you. This has been updated in the results section.

"Technical issues: Page2 Line72: please explain A&E"

Author response: Thank you. this has been updated and explained.

"References: the authors wrote double references (under numbers 8 and 9: Braley TJ,2010; and under numbers 36 and 57: Crank H,2017). Please check the other as well."

Author response: Thank you. The reference 9 was a typo and has been changed. Reference 57 has been removed.

Reviewer 2 Report

Authors were quite thorough in their analysis and the methodology was appropriately explained. The findings make an important contribution to the field. There were a few awkward sentences.

Author Response

"Authors were quite thorough in their analysis and the methodology was appropriately explained. The findings make an important contribution to the field. There were a few awkward sentences."

Author response: thank you for these comments. Sentences have been checked. 

Reviewer 3 Report

They should improve the summary, it is not a quantitative study, they should explain the study procedure in the systematic review.

They should review the methodology section, they should correctly include each information in each section (Databases, selection criteria, etc.)

They must improve the table 3, 4 and 5 to include the information in a more graphic way, there are unfilled lines.

Figure 2 must be its own, on the contrary, it should not be included if it does not have the appropriate permissions.

Author Response

"They should improve the summary, it is not a quantitative study, they should explain the study procedure in the systematic review."

Author response: thank you we have updated the abstract.

"They should review the methodology section, they should correctly include each information in each section (Databases, selection criteria, etc.)"

Author response: Thank you for this comment. We have updated this to follow the ENTREQ (Tong et al., 2012) for synthesis reviews.

"They must improve the table 3, 4 and 5 to include the information in a more graphic way, there are unfilled lines."

Author response: These tables were removed and the information they contained is now included in the result section.

"Figure 2 must be its own, on the contrary, it should not be included if it does not have the appropriate permissions."

Author response: This model was conceptualised by the authors and does not require permission. We have placed it with an explanation at the bottom of the results section as well as on the discussion section. 

Reviewer 4 Report

In this paper "A Thematic Synthesis Considering the Factors which Influence Multiple-Sclerosis Related Fatigue during Physical Activity", the authors have surveyed papers which talk about fatigue during exercise and have provided us with a systematic synthesis of several different factors that affect fatigue during exercise. The authors have done a very exhaustive search and refining process, which is clearly explained.  

While the majority of the manuscript talks about the several factors that affect fatigue during exercise which is part of the objective of this paper, the abstract also sets the goal of this manuscript to identify the necessary adaptations that may empower PwMS to remain active, which seems to be not that evident from the discussion and conclusion.

Most of the paper talks about the factors affecting fatigue during exercise for PwMS but doesn't mention the level of disability or severity of the patients considered in the studies discussed. It is a critical factor that must .be considered for fatigue-related studies in MS. As stated in line 691 by the authors, patients in different stages of the disease, and different level of disability and severity may have different experiences.

Although the manuscript talks about a systematic review process, it lacks "systematic" proofreading. There are several grammatical and punctuation errors that make some parts of the paper very difficult to read and comprehend. Below are some of the errors. However, I highly recommend the authors to proofread the manuscript for more of these errors. 

1. Line 45: because it is an invisible symptom, is difficult to be explained and measured.

The sentence appears to be missing a pronoun after the comma.

2. Line 57: Often as MS symptoms increase, PwMS acute incidents can impose hospitalization.

This sentence is unclear. Requires revision.

3. Line 59: NHS £46 million.

A new acronym is introduced here which hasn't been defined before.

4. Line 72: the rate of having first time A&E incidents.

A new acronym is introduced here which hasn't been defined before.

5. Line 94: identify how and at what extend PwMS are affected by MSRF,...

It is supposed to be "extent" and not "extend".

6. Line 154: to identify “fatally flawed” papers according the criteria were proposed by NHS...

Missing preposition after according. "were" is not required.

7. Line 179: In figure 1, the last box has a half complete "others = 6". Please add a proper figure.

These and several such errors are seen throughout the manuscript. Missing punctuations, spelling mistakes, and spelling mistakes are found all over the paper, making it very difficult to read.

Author Response

"While the majority of the manuscript talks about the several factors that affect fatigue during exercise which is part of the objective of this paper, the abstract also sets the goal of this manuscript to identify the necessary adaptations that may empower PwMS to remain active, which seems to be not that evident from the discussion and conclusion."

Author response: Thank you for the comment. We have added more information in the discussion section above Figure 2 and they are also fully presented in the results section.

"Most of the paper talks about the factors affecting fatigue during exercise for PwMS but doesn't mention the level of disability or severity of the patients considered in the studies discussed. It is a critical factor that must .be considered for fatigue-related studies in MS. As stated in line 691 by the authors, patients in different stages of the disease, and different level of disability and severity may have different experiences."

Author response: We have added a summary of the EDSS scores into the results section (see Table 1). We have added details in the supplementary file for the theme as identified above. We have also added this point into the limitations.

"Although the manuscript talks about a systematic review process, it lacks "systematic" proofreading. There are several grammatical and punctuation errors that make some parts of the paper very difficult to read and comprehend. Below are some of the errors. However, I highly recommend the authors to proofread the manuscript for more of these errors.

1. Line 45: because it is an invisible symptom, is difficult to be explained and measured.

The sentence appears to be missing a pronoun after the comma.

2. Line 57: Often as MS symptoms increase, PwMS acute incidents can impose hospitalization.

This sentence is unclear. Requires revision.

3. Line 59: NHS £46 million.

A new acronym is introduced here which hasn't been defined before.

4. Line 72: the rate of having first time A&E incidents.

A new acronym is introduced here which hasn't been defined before.

5. Line 94: identify how and at what extend PwMS are affected by MSRF,...

It is supposed to be "extent" and not "extend".

6. Line 154: to identify “fatally flawed” papers according the criteria were proposed by NHS...

Missing preposition after according. "were" is not required.

7. Line 179: In figure 1, the last box has a half complete "others = 6". Please add a proper figure.

These and several such errors are seen throughout the manuscript. Missing punctuations, spelling mistakes, and spelling mistakes are found all over the paper, making it very difficult to read."

Author response: Thank you for these comments. We have checked and corrected these and some other mistakes as well. 

Round 2

Reviewer 3 Report

I do not consider that an article consulting pages like google scholar or Researchgate is likely to be published in a scientific journal. This makes it lacking in rigor. Researchers should improve their methodology for carrying out a systematic review. A scientific review can not be done with pages without rigor or control.

Author Response

Reviewer: I do not consider that an article consulting pages like google scholar or Researchgate is likely to be published in a scientific journal. This makes it lacking in rigor. Researchers should improve their methodology for carrying out a systematic review. A scientific review can not be done with pages without rigor or control.

Author response: Thank you for this point.  However, Google Scholar has merit as part of a systematic review process.  For a long time, Google Scholar has been recognised as one mechanism of searching which is effective within a systematic review e.g., Shultz, M., “Comparing test searches in PubMed and Google Scholar”. JMIA 2007, 95, 442–445. And Anders, M. E. & Evans, D. P., “Comparison of PubMed and Google Scholar literature searches”. Respiratory Care, 2010, 55, 578–583.  The systematic review needs varied search methods like we have undertaken therefore, to remove it would be to illustrate a weakness of methodology.  Meanwhile, Research Gate can be removed from the manuscript without affecting the validity of the review methodology, because it was mainly used to access the research profiles of authors, as it has been mentioned on the next sentence of the paragraph in question.